# Liver Magnetic Resonance Elastography: Focus on Methodology, Technique, and Feasibility

**DOI:** 10.3390/diagnostics14040379

**Published:** 2024-02-09

**Authors:** Marta Zerunian, Benedetta Masci, Damiano Caruso, Francesco Pucciarelli, Michela Polici, Stefano Nardacci, Domenico De Santis, Elsa Iannicelli, Andrea Laghi

**Affiliations:** 1Department of Surgical and Medical Sciences and Translational Medicine, Sapienza University of Rome, Sant’Andrea University Hospital, Via di Grottarossa, 1035-1039, 00189 Rome, Italy; marta.zerunian@uniroma1.it (M.Z.); benedetta.masci@uniroma1.it (B.M.); michela.polici@uniroma1.it (M.P.); stefano.nardacci@uniroma1.it (S.N.); domenico.desantis@hotmail.it (D.D.S.); elsa.iannicelli@uniroma1.it (E.I.); andrea.laghi@uniroma1.it (A.L.); 2PhD School in Translational Medicine and Oncology, Department of Medical and Surgical Sciences and Translational Medicine, Faculty of Medicine and Psychology, Sapienza University of Rome, 00189 Rome, Italy

**Keywords:** magnetic resonance elastography, diffuse liver disease, imaging biomarker, fibrosis, magnetic resonance

## Abstract

Magnetic resonance elastography (MRE) is an imaging technique that combines low-frequency mechanical vibrations with magnetic resonance imaging to create visual maps and quantify liver parenchyma stiffness. As in recent years, diffuse liver diseases have become highly prevalent worldwide and could lead to a chronic condition with different stages of fibrosis. There is a strong necessity for a non-invasive, highly accurate, and standardised quantitative assessment to evaluate and manage patients with different stages of fibrosis from diagnosis to follow-up, as the actual reference standard for the diagnosis and staging of liver fibrosis is biopsy, an invasive method with possible peri-procedural complications and sampling errors. MRE could quantitatively evaluate liver stiffness, as it is a rapid and repeatable method with high specificity and sensitivity. MRE is based on the propagation of mechanical shear waves through the liver tissue that are directly proportional to the organ’s stiffness, expressed in kilopascals (kPa). To obtain a valid assessment of the real hepatic stiffness values, it is mandatory to obtain a high-quality examination. To understand the pearls and pitfalls of MRE, in this review, we describe our experience after one year of performing MRE from indications and patient preparation to acquisition, quality control, and image analysis.

## 1. Introduction

Magnetic resonance elastography (MRE) is an imaging technique that combines low-frequency mechanical vibrations with magnetic resonance imaging to create visual maps to quantify body tissue stiffness [1].

### 1.1. The Importance of Identifying Liver Cirrhosis

In recent years, MRE has been mainly used to evaluate liver parenchyma. As diffuse liver diseases are highly prevalent worldwide and could lead to a chronic condition of different stages of both fibrosis and cirrhosis, there is a strong necessity for a non-invasive, highly accurate, and standardised quantitative assessment to manage patients affected from early diagnosis to therapy management and follow-up [2]. The actual reference standard for the diagnosis and quantification of fibrosis in diffuse liver diseases is percutaneous biopsy. Even if the histology report has a high diagnostic value, this method has some limitations including low patient acceptance and has a 3% risk of complications such as pain, bleeding, or infection, with a mortality rate of 0.03% [3]; in addition, this procedure is also limited by the possibility of sampling errors due to the heterogeneous distribution of hepatic fibrosis and intra-operator variability [4,5].

### 1.2. Clinical Indications

Among the clinical indications to perform MRE, diffuse liver diseases that may lead to liver fibrosis are the most common. Diffuse liver diseases occur when the hepatocytes are damaged by various aetiologies, with consequent alterations of the hepatic tissue and possible fibrotic changes and cirrhotic alterations [6]. The most common among these diseases is non-alcoholic fatty liver disease (NAFLD), a highly diffused worldwide condition, particularly in Western countries, characterised by the accumulation of triglycerides within the hepatocyte, that owes its pathogenesis to the interaction between hormonal, nutritional, and genetic factors [7]. Important causes of diffuse liver diseases include alcohol-related liver disease, a condition associated with excessive alcohol consumption which can lead to both asymptomatic early steatosis to cirrhosis or hepatocellular carcinoma (HCC) [8], and viral hepatitis that can have an acute onset but could evolve into chronic forms and lead to different stages fibrosis. It is necessary to recognise that increased liver stiffness can be observed in patients without hepatic fibrosis; in particular, acute hepatic inflammatory processes, from infective or autoimmune causes, can diffusely increase hepatic stiffness [9]. Also, biliary obstruction, occurring mainly in the hilar region, can lead to elevated values of liver stiffness without the histologic presence of fibrosis [9]. Other causes of liver stiffness alterations, in the presence or absence of fibrotic alterations, include an intracellular overload of iron in hepatocytes, Ref. [6], or other substances including copper, or amyloid proteins as in primary or secondary amyloidosis [10]. Among other causes of diffuse liver alterations, there are the cardiovascular conditions mainly related to right heart failure, constrictive pericarditis, Budd–Chiari syndrome [11], veno-occlusive hepatic disease [12], or chronic cholestatic diseases [13]. All the above-mentioned conditions could evolve into hepatic cirrhosis, which is the end stage of mostly all the diseases cited. From a histological point of view, cirrhosis is characterised by hepatic fibrosis, due to the quantitative and qualitative changes in the liver extra-cellular matrix components, perfusion abnormalities, and nodular distortion of the hepatic architecture. The fibrotic changes can often appear as confluent fibrosis or bridging bands due to the excessive deposition of collagen fibres and consequent liver remodelling [14]. Cirrhosis is a chronic condition with a very socially high impact, considering that the incidence of new cases/year in Italy is around 30–60.000 new cases/year with a mortality rate that is luckily slowing down even if, in Europe, it is the fourth cause of death [12]. Hepatic fibrosis is a dynamic process that can be setback with appropriate therapy if detected early, or can progress to cirrhosis, causing hepatic failure and severe complications such as ascites, variceal bleeding, portal hypertension, or hepatocellular carcinoma [15]. As MRE has shown important results in early staging fibrosis in terms of specificity and sensitivity [16,17] with high levels of accuracy [18], it can play an important role in the multidisciplinary management of these conditions, from diagnosis to follow-up. For that reason, it is extremely important to perform this examination in a standardised manner, by following important practical steps to achieve a good quality of examination and thus analysis of the liver parenchyma. The main indication for performing MRE is detecting or staging hepatic fibrosis in patients with known fibrotic conditions or cirrhosis to quantify and grade the liver damage. Another important application of MRE is the assessment of fibrosis treatment response.

### 1.3. Imaging Methods to Identify Liver Fibrosis and Cirrhosis

Liver MRE has been shown to be an accurate method for detecting and staging liver fibrosis, especially in early stages, with a sensitivity, specificity, and organ coverage exceeding other techniques [15], such as the fibrosis serum marker test, ultrasound-based transient elastography, diffusion-weighted imaging, and computed tomography evaluation. Regarding ultrasound techniques, transient elastography (TE) is an imaging technique that allows rapid and non-invasive stiffness measurements, particularly helpful in patients with contraindications to MRI such as claustrophobia. However, TE has some limitations since the method does not explore the majority of the hepatic tissue and is associated with high failure rates in obese patients [19].

### 1.4. Diagnostic Accuracy of MRE to Identify Liver Fibrosis

Several studies have evaluated the accuracy of MRE in comparison with TE; a study performed by Imajo et al. [20] showed a higher accuracy of MRE in the diagnosis and staging of fibrosis and steatosis in NAFLD patients. A similar retrospective study performed by Park et al. showed similar results in terms of MRE and TE accuracy [19]. Regarding exclusively MRE, Wang et al. observed that MRE is able to identify fibrosis at stage ≥ 2 (F2-4) and stage ≥ 3 (F2-4), with a combination of high sensitivity (91%) and specificity (96%) [21].

### 1.5. Limitations and Contraindications of MRE

Despite the high accuracy and the non-invasiveness of the technique, some limitations need to be addressed, mainly the need for adequate patient preparation before the examination and the need for patient collaboration during MRE, since, for an accurate assessment to be possible, it is mandatory to have a satisfactory iconographic quality [22]. Also, MRE is associated with higher costs in comparison with other imaging techniques such as TE [23], and with the need for specific technical equipment, factors that reduce the availability of this imaging method. The other main limitations associated with MRE are related to the reduced performances in iron overload states or severe hepatic steatosis [5], and the difficulty of acquisition in severely obese patients. In addition, the presence of metallic joint protheses, artificial heart valves, a pacemaker, an implantable heart defibrillator, metal clips, cochlear implants, or any other type of metal fragments [24] are contraindications to perform MRE. Severe claustrophobia is also considered a contraindication to perform MRE [25].

### 1.6. Other Utilities and Future Applications of MRE

The utility of MRE as a tool to evaluate the organs’ stiffness has also been evaluated in other districts such as spleen [26], kidneys [27], and breast imaging [28]. Future applications in liver imaging include the characterisation of focal liver lesions and the possibility to distinguish between benign and malignant lesions [25], as the presence of focal liver lesions, benign or malignant, can determine focal increased liver stiffness [9].

## 2. Technical Aspects

### Basic Principles of MRE

MRE is an imaging technique based on the propagation of mechanical shear waves through the liver tissue to evaluate the organ’s stiffness. The shear wave generated during the examination has a velocity of propagation in the tissue which is directly proportional to the level of stiffness, expressed in kilopascals (kPa) [5]. Normal stiffness values are considered lower than 2.5 kPa [5,29]. MRE works by evaluating the motion of the mechanical shear waves in the liver using motion-sensitised sequences and then inverting the wave pattern to obtain tissue stiffness values.

The shear waves are generated by a mechanical wave driver connected to a passive pneumatical driver placed over the right lower chest wall of the patient in a supine position [3]. A continuous acoustic vibration is generated and transmitted to the liver via the passive driver. The mechanical waves used in MRE are typically in the acoustic frequency range of 40–150 Hertz (Hz). Axial images of the waves’ propagation in the liver are acquired with gradient-recalled echo (GRE) pulse sequences synchronised to the frequency of the vibrations applied to the liver. Images are acquired during consecutive breath holds of 10–15 s each [25]. Generally, a minimum number of 4 slices is required [29] to obtain significative liver quantitative measurements. MRE is also available in different pulse sequences other than 2D GRE, such as 2D spin-echo (SE) echo-planar imaging and three-dimensional (3D) SE echo-planar imaging [30]. Regarding 2D SE echo-planar imaging MRE sequences, recent studies have shown a shorter scan time and higher signal-to-noise ratio; also, 2D SE echo-planar imaging has the capability to achieve images with a single breath-hold scan and provides larger measurable ROIs with a lower failure rate [31]. At the moment, not all the scanners have the possibility to perform both 2DGRE and 2DSE depending on the indications, as well as for 3D SE; despite the advantages of the latter two sequences as described above, some scanners still have the possibility of 2D GRE acquisition only for clinical use. For these reasons, we describe our experience with 2D GRE according to our routine usage of MRE. After the scanning, from a single acquisition, we obtain the following: a magnitude image which gives anatomic information, a phase contrast image which gives wave motion information, and both a greyscale and colour elastogram, with and without a superimposed 95% confidence map. The confidence map is a statistical derivation of the stiffness map used to remove the liver regions with less authentic values of stiffness data due to the presence of artifacts or discontinuous propagation of the waves. Before the end of the acquisition, it is mandatory to evaluate the quality of the examination in order to obtain a valid assessment of the most reliable values of hepatic stiffness. The considered imaging criteria of satisfactory image quality are the evaluation of liver motion in the magnitude image, the presence of the characteristic “signal void” in the abdominal tissues below the passive driver, and the presence of visible liver waves in the wave images.

## 3. How We Do It

### 3.1. Patient Preparation

Adequate preparation before performing MRE is essential to obtain an accurate, repeatable, and successful exam.

The patient’s preparation for MRE is similar to the preparation for a standard upper abdomen MRI examination; the patient should be in fasting status for at least 4 to 6 h prior to the examination. In fact, even if the technique does not require contrast medium injection, post-prandial status could increase hepatic stiffness in patients with chronic liver conditions [5] and determine changes in stiffness values. To provide reproducible measurements, the same preparation is needed for follow-up examinations. Another important aspect, related to the mechanical wave propagation, is to undergo this examination with an empty stomach to avoid artifacts. Before the examination, it is mandatory to obtain informed consent from the patient and to explain how the examination will be performed to the subject, to improve collaboration between all the professional figures included in MRE and the patient. In fact, in our experience, we observed that it is incredibly helpful to show the passive pad to the patients before the beginning of the examination to familiarise the patient with the device while in a neutral zone, not in the MR scanner which could be a stressful location for the patient. In addition, it is fundamental that the operator explains to the patient that this mechanical wave will be felt during the scan, at end-expiration. This will allow for respiratory acts appropriate to the scan in order to reduce artefacts and abrupt movements of the patient, which could invalidate the examination.

### 3.2. Technical Details of MR Scan

In our experience, we performed MRE on a 1.5 T scanner (Signa Voyager, GE Healthcare, Waukesha, WI, USA) with acquisitions of 2D axial gradient-recalled echo (GRE) sequences (MR-Touch^®^ GE Healthcare, Milwaukee, WI, USA).

The sequence was acquired with a breath-hold technique of 14 s for each of four slices obtained at end-expiration; multiple studies have found a significantly higher reproducibility of MRE when a breath hold is performed at end-expiration [32]. Details of the acquisitions’ parameters are shown in Table 1.

The hardware components that induce vibrations into the subject are an active pneumatic mechanical wave driver and a rigid passive driver. The active driver is located outside the scan room, as placing the active driver near the MR scan is not recommended for it can be noisy when running; the latter is connected to the passive driver with a rigid plastic tube. The passive driver is a rigid, plate-like device that is fastened to the abdominal wall with an elastic strap; it generates a continuous acoustic vibration at a fixed frequency, that is transmitted through the abdomen during the entire examination.

## 4. Performing MRE Step-by-Step

### 4.1. Passive Driver Positioning

First, the passive driver needs to be connected with a rigid plastic tube that vehicles the mechanical wave from the external air compressor. For the best MRE examination technique, the rigid plastic tube has its own convexity that needs to be maintained during the passive driver positioning and during the scanning, as reported in Figure 1.

Following these simple tips will optimise patient comfort and consequently the acquisition and subsequent data processing. MRE: magnetic resonance elastography.

Then, the passive driver should be placed over the right hepatic lobe, in order to perform MRE over the largest portion of the hepatic tissue. To localise the right hepatic lobe, the sternum xiphoid process and the right mid clavicular line are used as anatomical landmarks. As the pad positioning may need to be adjusted for some patients depending on individual anatomy, before the examination, the radiologist usually performs an end-expiration supine abdominal palpation to evaluate where to place the pad. The passive driver is then tightly secured with an elastic strap around the patient’s torso, with the patient at end-expiration, as maintaining adequate contact between the abdominal wall and the pad is fundamental to improve the quality of the examination. Even though the pad should be tightly fastened, especially for obese patients, it is important to still allow the subject to breath comfortably. For small patients, a thin cloth is usually used to eliminate the air interposition between the pad and abdominal wall.

### 4.2. Abdominal Coil

To improve patients’ comfort during the examination and to reduce the sense of bulk on the abdomen, we use a very light and flexible coil with 32 channels with Adaptive Image Receive (AIR™, GE Healthcare, Milwaukee, WI, USA radiofrequency technology. This coil allows us to adapt it to the patient’s anatomy, without compromising the passive driver placement, and to gain signal thanks to the high number of channels than usual abdominal rigid coils.

### 4.3. Slices’ Positioning

Once the passive driver is positioned and connected to the active driver outside the scan room, the localiser scan is obtained; the localiser scan should be obtained at end-expiration to determine both the positioning of the four slices of MRE and the pad localisation, visible as a flattening of the abdominal wall. The slices should include the liver’s largest portion, avoiding the inferior portions and the dome, as these areas suffer from more respiratory artifacts. In fact, slice positioning on this area may result in focal, elevated liver stiffness on colour elastograms due to the waves’ orientation as they pass obliquely to the liver dome due to the shape of the organ; oblique waves will appear thicker than waves that transversely pass through the organ, mimicking elevated liver stiffness values [5].

### 4.4. Acquiring the Images

Once the localiser scan images are obtained and the four slices are adequately placed, the examination can start. Each of the four sets of images are acquired with a GRE pulse sequence with serial breath holds of 14 s each; these sequences have special motion-encoding gradients synchronised to the frequency of the vibrations applied to the liver. It is important in this step that the radiographer is well trained to guide the patients through the breathing cycle and also that the radiographer checks the respiratory patients’ movement from the internal camera before the start of the acquisition. In our experience, we did not perform MRE after the administration of contrast medium, as different study results have shown no significant differences in stiffness values when performing MRE before and after contrast medium administration [5,33].

### 4.5. Passive Driver Vibration and Amplitude

We performed MRE with stiffness thresholds established at 60 Hz, as shown in Table 1; it is mandatory to perform follow-up examinations at the same frequency of the baseline examination as liver stiffness measurements are frequency-dependent. The mechanical waves used in MRE are usually in the acoustic frequency range of 40–150 Hz and studies are prevalently performed with continuous vibrations at 50 Hz, 60 Hz, or 80 Hz [34]. The passive driver amplitude, which determines the intensity of the shear waves applied to the liver, is set at 90%; even though it can be modified depending on the patient’s body habitus (i.e., paediatric patients), in our experience, we only performed MRE on adult patients, at 90% amplitude with optimal results in terms of quality of the examination.

### 4.6. Quality Control

As mentioned before, not only is the image acquisition step important to obtain valid liver stiffness measurements; in fact, after the image acquisition, MRE needs to be post-processed with an inversion algorithm on a dedicated workstation. Post-processing includes the extraction from qualitative colour stiffness maps and greyscale stiffness maps. For the abovementioned, when acquired, each image obtained should be evaluated to ensure their quality and, eventually, repeat the acquisition (Figure 1). The technician should check the magnitude image to evaluate the imaging motion and the presence of respiratory artifacts and check the presence of a “signal void” in the subcutaneous abdominal tissues directly below the passive driver. The presence of the signal void is mandatory and its absence needs to be considered as non-propagation waves. Then, the technician should review the phase images to evaluate the waves’ propagation in the liver and the reconstructed wave images, typically displayed in colour, to exclude wave distortion or low-quality wave propagation. High-quality wave propagation can be recognised as waves will form parallel to the liver surface and propagate homogeneously through the entire organ. Poor-quality wave propagation is characterised by waves that are not parallel to the liver surface, that are low-amplitude, and that propagate unevenly throughout the liver.

The last step of the quality control performed by the technician is evaluating both the greyscale and colour elastograms; an elastogram is considered of high quality when a large area of the organ, visually and objectively assessed by placing regions of interest (ROIs) covering a total amount of at least 500 pixels, is not covered by the 95% confidence map, so the largest portion of the liver can be evaluated.

### 4.7. Image Analysis

After a good-quality examination is assured, ROIs can be drawn to evaluate liver stiffness measurements. Both manual and automated measurements can be performed; in our experience, we have drawn free-hand ROIs on colour stiffness maps including the largest volume possible of the liver parenchyma, with magnitude images as optional anatomical references. ROI placing and liver stiffness measurements are generally performed on a dedicated workstation. As suggested by Guglielmo et al. [5], we excluded the following from ROIs: wave artifacts, large hepatic vessels, narrow liver segments, and the left liver lobe due to heart beat artifacts. Also, we avoided drawing ROIs in the gallbladder fossa and at the liver edge. To consider MRE as diagnostic, as previously mentioned, the total amount of pixels included in all ROIs should be at least 500. The weighted arithmetic mean is then calculated, considering the area sampled and the liver stiffness of each slice with the formula shown:Weighted arithmetic mean = (*m*1*w*1 + *m*2*w*2 + *m*3*w*3 + *m*4*w*4) ÷ (*w*1 + *w*2 + *w*3 + *w*4) 
where *m*1, *m*2, *m*3, and *m*4 are the mean stiffness values of each area sampled, and *w*1, *w*2, *w*3, and *w*4 are the ROI sizes in pixels per cm^2^. The weighted arithmetic mean can then be used to correlate with fibrosis stage; examples are provided in Figure 2 and Figure 3.

In particular, values of hepatic stiffness, assessed at 60 Hz, lower than 2.5 kPa are considered normal, 2.5–2.9 kPa are considered normal or compatible with inflammation, values of 2.9–3.5 kPa are compatible with Stage 1 to 2 of fibrosis according to METAVIR scoring [35], 3.5–4 kPa are considered Stage 2 to 3 fibrosis [35], values between 4 and 5 kPa are classified as Stage 3 to 4 fibrosis, and values higher than 5 kPa are considered as Stage 4 fibrosis or cirrhosis [35,36].

### 4.8. Frequently Observed Pitfalls

In our experience, we faced different causes of low-quality or non-diagnostic elastograms. Common issues of low-quality elastograms include technical issues such as active driver failure; the passive driver being inadequately secured to the abdominal wall, especially in obese patients; a disconnected or kinked tube connecting the active and passive driver; or poor slice positioning, as described in Table 2.

When performing MRE in obese patients, strategies aimed at increasing pad adherence (i.e., placement of additional elastic strap) might reduce technical failures; in addition, simple technical failures can be avoided in a trained team by checking the tube positioning before starting each examination. Other causes of failed elastograms were observed in the literature; Guglielmo et al. [5] described the following as important causes of low-quality elastograms: liver parenchymal diseases such as significant iron overload or severe steatosis. Iron overload generally results in a lower liver signal-to-noise ratio, with a non-diagnostic elastogram, as shown in Figure 4.

### 4.9. Accuracy of MRE

MRE is considered an accurate and effective tool to state liver stiffness in fibrotic and non-fibrotic liver. A retrospective study performed by Venkatesh et al. [17] stated an MRE accuracy in detecting liver fibrosis in chronic HBV patients in comparison with serum biomarkers, with a significantly higher accuracy of MRE (0.99 vs. 0.55–0.73). Another retrospective study performed by Chen et al. [18] assessed the accuracy of MRE in differentiating NASH from steatosis (Area Under Curve = 0.93). A retrospective study from Morisaka et al. [41] evaluated the accuracy of MRE in comparison for the diagnosis of different grades of fibrosis in a retrospective study, obtaining statistically equivalent results between the two methods. Finally, Loomba et al. [42] showed, in a prospective study, how MRE has a high accuracy in predicting advanced fibrosis in NAFLD by discriminating stage 0–2 from stage 3–4 of fibrosis (Area Under Curve = 0.924). A recent meta-analysis from Selavaraj E.A. et al. [43] analysed the diagnostic accuracy of MRE in NAFLD patients, showing a sensitivity, specificity, and summary AUC (sAUC) for diagnosing fibrosis stage ≥ F1 of 71%, 85%, and 0.87; the diagnostic accuracy in detecting significant (≥F2) and advanced fibrosis (≥F3) and cirrhosis (F4) showed an sAUC, sensitivity, and specificity of 0.91, 78%, and 89%, and 0.92, 83%, and 89%, respectively. Also, the same meta-analysis showed a diagnostic accuracy in detecting liver cirrhosis with an sAUC, sensitivity, and specificity of 0.90, 81%, and 90%, respectively. A meta-analysis by Schambek J.P.L et al. [44] analysed the diagnostic performance of MRE and point shear wave elastography (pSWE) with an excellent diagnostic accuracy of MRE in staging METAVIR F2 patients with a pooled sensitivity and specificity of 0.94 and 0.95 in MRE, higher than the pooled sensitivity and specificity of pSWE. In this meta-analysis, MRE proved to be an important tool for the early diagnosis of liver fibrosis, reducing the role of biopsies in F2 fibrosis patients.

## 5. Conclusions

In conclusion, MRE can be considered a valid tool to evaluate liver stiffness; it is non-invasive, relatively simple to perform once the technical team is trained, repeatable, and capable of recognising staging fibrosis early with high levels of sensitivity and specificity. Also, MRE is considered an imaging method with a higher reliability and reproducibility in comparison with other imaging modalities, such as TE [14]. MRE is particularly useful to evaluate the response to therapies, as it is a non-operator-dependent method and is preferable to TE in severe obese patients, patients with narrow intercostal spaces, and/or ascites, with a lower failure rate. Future directions of MRE include the differentiation between liver stiffness caused by fibrosis, oedema, inflammation, and passive congestion, to assess precocious liver manifestations in the context of alcohol consumption or in diabetes, to screen drug-mediated liver damages, or to monitor the response in locoregional treatments for hepatocellular carcinoma (HCC) [29]. The increasing spread of this technique in clinical practice will increase clinical interest as a dedicated non-invasive tool to support medical decisions in chronic disease. In addition, technical implementations of such 3D sequences and more comfortable hardware will help to increasingly refine and standardise the technique for use in clinical practice.

## Figures and Tables

**Figure 1 diagnostics-14-00379-f001:**
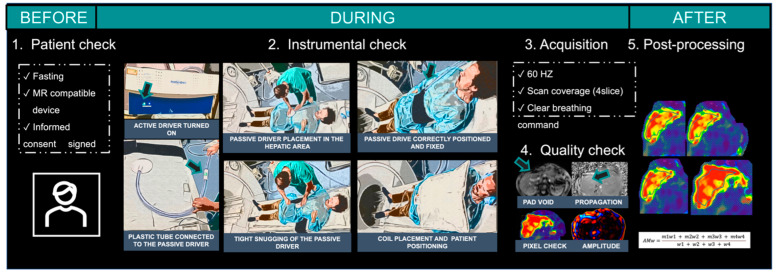
Some important steps before (1. Patient check), during (2. Instrumental check, 3. Acquisition, and 4. Quality check), and after (5. Post-processing) MRE acquisition.

**Figure 2 diagnostics-14-00379-f002:**
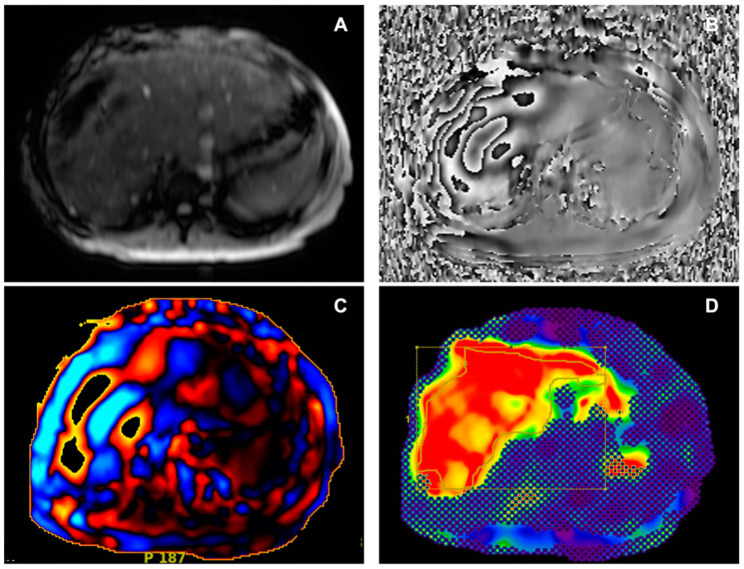
Female, 33 years old, with NASH and hepatic fibrosis (7.4 kPa). (**A**) Axial magnitude image showing the signal void in the right anterior abdominal subcutaneous tissues. (**B**) Phase image showing waves propagating through liver tissue. (**C**) Wave image showing wave propagation, with waves moving parallel to the liver surface, thicker than those in non-fibrotic liver. (**D**) Corresponding colour elastogram, with free-hand ROI placed in the liver tissue not covered by the 95% confidence map; the colours red and orange are associated with elevated stiffness values. NASH: non-alcoholic steatohepatitis; kPa: kilopascals; ROI: region of interest.

**Figure 3 diagnostics-14-00379-f003:**
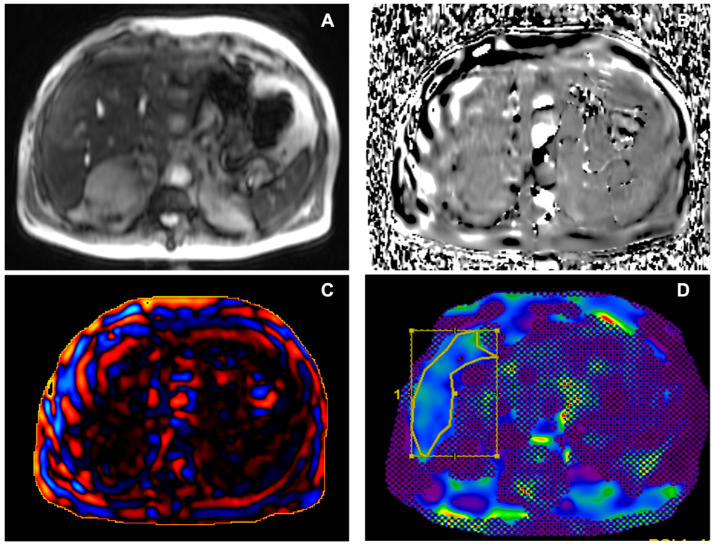
Male, 69 years old, with multiple metabolic risk factors and l grade hepatic fibrosis (2.9 kPa). (**A**) Axial magnitude image showing the signal void in the anterior abdominal subcutaneous tissues. (**B**) Phase image showing waves propagating through liver tissue. (**C**) Wave image showing wave propagation, with thin waves moving parallel to the liver surface. (**D**) Corresponding colour elastogram, with free-hand ROI placed in the liver tissue not covered by the 95% confidence map; the colour blue is generally found in non-fibrotic or low-grade fibrosis stiffness values. kPa: kilopascals; ROI: region of interest.

**Figure 4 diagnostics-14-00379-f004:**
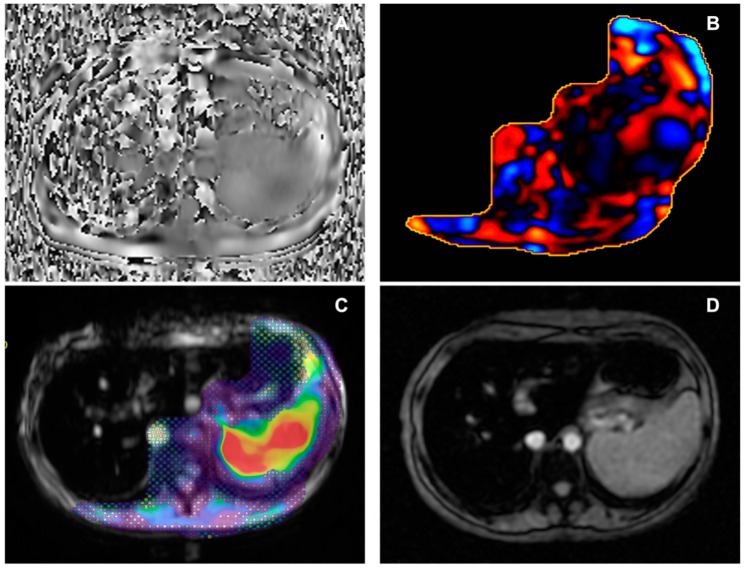
Male, 38 years old, with severe iron overload. (**A**) Axial phase image showing the absence of wave propagation through the liver. (**B**) Non-diagnostic wave image. (**C**) Elastogram reconstruction with the interposition of the 95% confidence map not covering the liver area. (**D**) Axial Starmap T2* breath-hold sequence, confirming iron overload. In the case of severe iron overload, an option might be to use SE echoplanar sequences instead of GRE sequences, with the latter being more prone to non-diagnostic MRE, even if an univocal cut-off of severe iron overload of MRE failure has not been established yet [37,38]. As observed by a retrospective study by Meng Y. et al. [39] on 1377 consecutive MRE examinations, MRE often had a low failure rate (5.6%), with the majority of failure cases due to inadequate signal-to-noise ratio related to iron overload (3%) and the remaining cases caused by execution errors or respiratory artifacts. A retrospective study by Wagner M. et al. [40] confirmed the higher rate of technical failure associated with liver iron overload with *p* < 0.001. Also, the presence of bowel interposition between the liver and abdominal wall, motion artifacts, or interfering paramagnetic materials are considered as causes of low-quality elastograms. Despite the lack of published data on the subject from our centre, the failure rate aligns with literature data after the initial eight weeks of training.

**Table 1 diagnostics-14-00379-t001:** Imaging parameters of the sequence used in our clinical practice.

	Imaging Parameters 2D GRE 1.5 T		Driver Parameters
Pulse sequence	MR-Touch	Amplitude (%)	90
Bandwidth (kHz)	62.50	Frequency (Hz)	60
FOV (mm)	430 × 430		
Matrix	224 × 64		
TE (msec)	Minimum TE (~20)		
TR(msec)	50		
Flip angle (°)	30		
NEX	1		
Fat suppression	active		
Slice thickness (mm)	10		
Slice gap (mm)	0		
N. of slices	4		

GRE: gradient-recalled echo; T: Tesla; Hz: hertz; kHz: kilohertz; FOV: field of view; TE: echo time; msec: milliseconds; TR: repetition time; NEX: number of excitations; mm: millimetres.

**Table 2 diagnostics-14-00379-t002:** Possible causes of MRE failure checklist.

Technical Issues	Liver Diseases/Anatomical Conditions
Active driver failure	Liver iron overload (70%)
Incorrectly positioned/secured passive driver	Severe steatosis
Improper slice selection	
Disconnected/Kinked tube connecting active to passive driver	
Poor breath-hold	

MRE: magnetic resonance elastography.

## Data Availability

The material used for this review article is available from the corresponding author on reasonable request.

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
