# Peer review of "Liver Magnetic Resonance Elastography: Focus on Methodology, Technique, and Feasibility"

_diagnostics, 2024, doi:10.3390/diagnostics14040379_

Round 1

Reviewer 1 Report

Comments and Suggestions for Authors

The study by Zerunian M et al is a narrative review regarding the methodology of liver magnetic resonance elastography (MRE). However, authors should make important changes to improve their narrative review’s quality and be accepted in diagnostics.

Major comments

1.     Title. This review is focused on the “methodology” of MRE and the sections about applications and clinical indications of MRE are very limited. Therefore, the title should be changed and simplify 

2.     Introduction. Please define cirrhosis from a histological point of view (line 80). Please reorganize the introduction into different paragraphs according to the different subjects such as 1) the importance of identifying cirrhosis, 2) methods to identify cirrhosis, 3) Technical characteristics of MRE, 4) Diagnostic accuracy to identify cirrhosis, 5) Limitations, and 6) other utilities of MRE.

3.     Technical aspects. Please describe how to evaluate the quality of the examination and the criteria, and the rate of excluded examination due to suboptimal criteria

4.     Table 1. Please include a footnote explaining each parameter

5.     Figure 1. Please include the number of the different sections in Figure 1  and include in the paragraph the reference to the figure

6.     Quality control. Please define "large area" (line 257)

7.     Please include METAVIR as the fibrosis stage classification and the reference (lines 296)

8.     Pitfalls. Please include the frequency of these low-quality elastograms that can reduce the applicability of these techniques and discuss it as a limitation. 

9.     Please discuss the overall limitations according to MR, and according to MRE (preparation, obtention, and postproduction)

10.  Accuracy of MRE. The section is minimal. Authors should include more actualized references, prospective studies, reviews,  and meta-analyses.

Reviewer 2 Report

Comments and Suggestions for Authors

The paper is very interesting, describing very well the technique. 

The only drawback is that you do not refer the major limitations: the high costs, the need of patients' help to have a good quality exam and the lack of availability of the exam. You should focus on the conclusions the specific situations for this type of exam comparing to transient elastography (when MRE is clearly better than TE)

Round 2

Reviewer 1 Report

Comments and Suggestions for Authors

The authors have made modifications to the original manuscript based on the reviewers’ comments and advice improving the quality of their study.

Now, the manuscript is suitable for publication in diagnostics